# Predictive value of cystatin C and neutrophil gelatinase-associated lipocalin in contrast-induced nephropathy: A meta-analysis

Yi He, Yunzhen Deng[◉], Kaiting Zhuang[◉], Siyao Li[◉], Jing Xi[◉], Junxiang Chen[◉]*

Department of Nephrology, Hunan Key Laboratory of Kidney Disease and Blood Purification, The Second Xiangya Hospital of Central South University, Changsha, Hunan, China

◉ These authors contributed equally to this work.
* chenjxly@csu.edu.cn

## Abstract

### Background

There are still limited studies comprehensively examining the diagnostic performance of neutrophil gelatinase-associated lipocalin (NGAL) and cystatin C in contrast-induced nephropathy (CIN). The study aimed to investigate and compare the predictive value of NGAL and cystatin C in the early diagnosis of CIN.

### Methods and materials

We searched the PubMed, EMBASE and Cochrane Library databases until November 10, 2019. The methodological quality of the included studies was assessed by the Quality Assessment of Diagnostic Accuracy Studies 2 (QUADAS-2) tool. Bivariate modeling and hierarchical summary receiver operating characteristic (HSROC) modeling were performed to summarize and compare the diagnostic performance of blood/urine NGAL and serum cystatin C in CIN. Subgroup and meta-regression analyses were performed according to the study and patient characteristics.

### Results

Thirty-seven studies from thirty-one original studies were included (blood NGAL, 1840 patients in 9 studies; urine NGAL, 1701 patients in 10 studies; serum cystatin C, 5509 patients in 18 studies). Overall, serum cystatin C performed better than serum/urine NGAL (pooled DOR: 43 (95%CI: 12–152); AUROC: 0.93; λ: 3.79); serum and urine NGAL had a similar diagnostic performance (pooled DOR: 25 (95%CI: 6–108)/22(95%CI: 8–64); AUROC: 0.90/0.89; λ: 3.20/3.08). Meta-regression analysis indicated that the sources of heterogeneity might be CIN definition, assays, and nationalities.

### Conclusion

Both NGAL and cystatin C can serve as early diagnostic indicators of CIN, while cystatin C may perform better than NGAL.

**Data Availability Statement:** All relevant data are within the paper and its Supporting Information files.

                                                                                                                  

**Funding:** This work was funded by the National Natural Science Foundation of China (Grant No. 81770692), and JX Chen was the author who received support. There is no commercial interference. Besides, the funders had no role in study design, data collection and analysis, decision to publish, or preparation of the manuscript.

**Competing interests:** The authors have declared that no competing interests exist.

## Introduction

Contrast-induced nephropathy (CIN) is defined as acute kidney injury (AKI) occurring 24–72 h after radiographic contrast media (CM) exposure in the absence of an alternative etiology[1]. After decreased renal perfusion (42%) and postoperative acute renal failure (18%), CIN is the third most common cause (12%) of hospital-acquired kidney failure[2, 3]. Half of the patients who develop CIN undergo cardiac catheterization and percutaneous coronary intervention (PCI)[2, 4]. CIN has become a major healthcare issue and is associated with adverse events, length of hospital stay and healthcare cost[5].

Currently, the diagnosis of CIN is based on the variation in serum creatinine (sCr) levels before and after CM exposure. However, sCr is a delayed and not always reliable indicator. After the kidneys undergo a contrast-induced toxicity attack, sCr typically increases within the first 24–48 h, peaks at 3–5 days and returns near baseline within 1–3 weeks[6]. The change in sCr is not evident until 50% of the nephrons have already been injured[7]. Furthermore, sCr can vary with many factors, such as age, sex, muscle mass, muscle metabolism, medications and hydration status[8]. Since there are so many limitations of sCr, the urgency for finding specific and sensitive biomarkers is highlighted. Besides, desirable biomarkers should also be rapidly quantifiable for analysis, which allows timely clinical interventions to be made[5].

Several promising biomarkers have been identified for the early diagnosis of CIN[1, 5]. Among them, neutrophil gelatinase-associated lipocalin (NGAL) and cystatin C are the most frequently investigated biomarkers in the clinic. NGAL is a 25-kDa protein covalently bound to gelatinase from secondary granules of human neutrophils and can reflect the damage of tubule cells[9, 10]. As the earliest biomarker after kidney injury, NGAL can be secreted and released into blood and urine in a short time and strongly correlates with sCr levels for CIN diagnosis[11]. After CM exposure, the levels of serum and urinary NGAL rise within 2 and 4 hours, respectively[12, 13]. Cystatin C is a 13-kDa cysteine proteinase inhibitor produced by nucleated cells and can be freely filtered by glomeruli, then reabsorbed and catabolized by the tubular cells[14]. It is less influenced by age, sex, race, muscle mass, steroid therapy, infection, liver disease or inflammation[15]. As cystatin C is merely distributed in the extracellular fluid volume and has a smaller distribution range than that of creatinine, serum cystatin C rises more rapidly than serum creatinine when GFR decreases[16–18]. Thus, serum cystatin C is a more accurate and earlier marker of GFR reduction than sCr.

Currently, multiple studies have reported that either NGAL or cystatin C alone could be viewed as a valuable predictor of early diagnosis for CIN; however, the comparison of the diagnostic performance between NGAL and cystatin C is still controversial and limited. Thus, we systemically reviewed relevant references and conducted a meta-analysis to summarize the predictive ability of serum/urine NGAL and serum cystatin C and to further compare those indicators on different occasions in order to provide significant evidence for the early diagnosis of CIN, which may provide more benefits for timely intervention and improvement of prognosis.

## Methods and materials

This systematic review and meta-analysis was performed according to the Cochrane Handbook for Systematic Reviews of Diagnostic Test Accuracy and reported according to the Preferred Reporting Items for Systematic Reviews and Meta-Analyses of Diagnostic Test Accuracy Studies (PRISMA-DTA) statement[19].

Two independent investigators (Yi He and Yunzhen Deng) conducted the "Data source" "Study selection" and "Data extraction and quality assessment" parts separately, and any disagreements were solved by discussion.

## Data source

PubMed, EMBASE and Cochrane Library databases were searched to identify possible references up to November 10, 2019. The electrical search strategy was developed based on the PICO format (P, patients/participants/population; I, index tests; C, comparator/reference tests; O, outcome), and search keywords were established using MeSH forms (PubMed) and Emtree forms (EMBASE).

The search terms are displayed as follows (Table 1).

## Study selection

Raw data from separate databases were pooled in EndNote (version X9, Thomason Reuters Company) and screened to identify eligible studies. Duplicate records were removed. The inclusion and exclusion criteria were as follows.

## Inclusion criteria

1. Original clinical articles with adult participants (no restriction on prospective or retrospective studies).

2. Patients with suspected CIN or contrast-induced acute kidney injury.

3. NGAL (serum, plasma or urine source) or cystatin C performed as index tests.

4. Sufficient information to reconstruct a 2×2 table (sample capacity, sensitivity, and specificity, etc.).

## Exclusion criteria

1. Irrelevant article types: case reports, letters, replies, editorials, guidelines, consensus, conference abstracts, reviews, meta-analyses, or clinical trials.

2. Animal experiments.

3. Only reported the correlation between biomarkers and CIN/contrast-induced acute kidney injury.

## Data extraction and quality assessment

The characteristics and outcome data of the eligible studies were extracted according to the standardized form. The extracted data included study characteristics (first author, publication

Table 1. The search terms used in systematic review.

| Frame | Search terms Diagnostic accuracy of NGAL versus cystatin C in contrast-induced nephropathy |
| --- | --- |
| Population | (contrast media) or (contrast agent) or (contrast materials) or (contrast material) or (radiocontrast media) or (radiocontrast agent) or (radiocontrast agents) or (radiopaque media) |
| Index tests | Lipocalin-2 or Lipocalin2 or (NGAL protein) or (oncogene 24p3 protein) or (siderocalin protein) or (neutrophil Gelatinase-Associated Lipocalin) or (neutrophil Gelatinase Associated Lipocalin) or (Lipocalin-2 protein) or (Lipocalin 2 protein) |
| Comparator | (Cystatin C) or (post-gamma-Globulin) or (post gamma Globulin) or (Neuroendocrine Basic Polypeptide) or (Cystatin 3) or (gamma-Trace) or (gamma Trace) |
| Outcome | (acute kidney injury) or (acute kidney injuries) or (acute renal injury) or (acute renal injuries) or (acute renal insufficiency) or (acute renal insufficiencies) or (acute kidney insufficiency) or (acute kidney failure) or (acute kidney failures) or (acute renal failure) or (acute renal failures) or (kidney disease) or (kidney diseases) or nephropathy |

year, study region, study design, and CIN/contrast-induced AKI definition), patient characteristics (number of patients, age, sex distribution, baseline renal function, and settings) and index test characteristics (detection method of index, evaluation time, sample source, and cut-off value). A 2×2 table was constructed according to the study outcomes (true-positive (TP); true-negative (TN); false-positive (FP); and false-negative (FN) results). If only sensitivity and specificity were displayed in eligible studies, the 2×2 table would be created via the Bayesian method, with which the outcome data being back-calculated according to the sample capacity.

The methodological quality of the eligible studies was assessed by the Quality Assessment of Diagnostic Accuracy Studies 2 (QUADAS-2) tool[20]. The methodological quality graph and methodological quality summary were conducted by Review Manager (version 5.2. Copenhagen: The Nordic Cochrane Centre, The Cochrane Collaboration, 2012).

## Statistical analysis

The statistical analyses were performed using the "midas" and "metandi" modules in Stata software (version 14.2; StataCorp LP, College Station, TX) and Review Manager 5.2.

A mixed bivariate random-effects model was used for analyzing and pooling the diagnostic accuracy measurements across studies. We plotted the summary estimates of each test in forest plots and hierarchical summary receiver operating characteristic (HSROC) curves. The summary results are displayed as the 95% confidence region and 95% prediction region in the HSROC curve plot.

Heterogeneity was detected using the Cochrane Q test (P<0.05 indicates the presence of heterogeneity) and Higgins' $I^2$ test (heterogeneity can be roughly evaluated according to the value of $I^2$ as follows: $I^2$: 0–25%, might not be important; 25–50%, low heterogeneity; 50–75%, moderate heterogeneity; 75%-90%, high heterogeneity)[21]. The source of heterogeneity from the threshold effect can be assessed in three ways. The first way is to check the coupled forest plot of sensitivity and specificity (an inverse change in the side-by-side display of sensitivity and specificity in the forest plot indicates the presence of a threshold effect). The second way is to calculate the Spearman correlation coefficient between the sensitivity and false-positive rate (a coefficient >0.6 indicates a considerable threshold effect). The third way is to draw an SROC plot. The points in the plot showed an overall curvilinear distribution (from the lower-left corner to the upper right corner) in the ROC space, indicating the presence of a threshold effect[22].

Sensitivity analysis was conducted to examine the stability by omitting each study at a time to eliminate factors that influence heterogeneity. Meta-regression analyses using several covariates were conducted to explore the source of heterogeneity without the threshold effect.

Deeks funnel plot was performed to evaluate the publication bias (P value<0.1 indicates the presence of publication bias).

## Results

### Literature search and selection

A diagram of the literature search and selection process is presented in Fig 1. The initial search from three databases identified 1450 relevant records. After removing 306 duplicate records, 1144 references were screened by title and abstract. A total of 1036 records were excluded for irrelevant article types and irrelevant content, e.g., randomized controlled trials and animal experiments. For the remaining 108 studies, the full texts were further assessed according to the inclusion criteria, and articles that contained pediatric patients, insufficient evidence or only reported correlations were removed. Finally, a total of 32 studies including 9088 patients were included in the quality assessment.

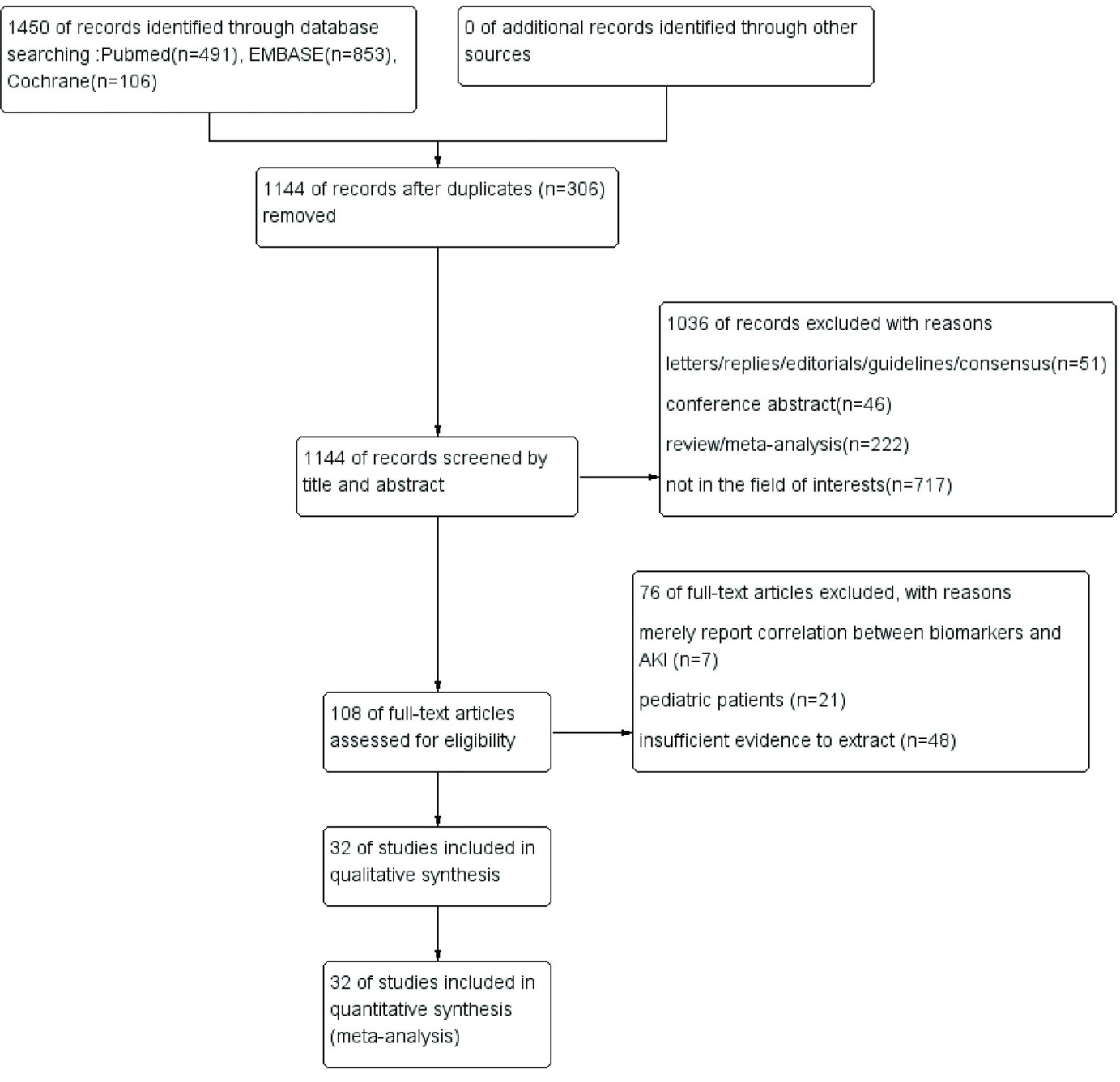

**Fig 1. The process of study search and selection.**

## Characteristics of the included studies

The included study characteristics, demographic features, and index test characteristics are summarized in Table 2 and Table 3.

Most of the included studies were prospective studies (n = 30), while one of the remaining studies was a retrospective study and the other was a nested case-control study. Among them, most studies were performed in patients undergoing percutaneous coronary intervention (PCI)/coronary angiography (CAG). The diagnostic performance of blood NGAL, urine

**Table 2. The characteristics of included study and population.**

| First Author | Year | Location | Study design | CIN definition | No. of patient | No. of CIN | Mean age[a] | Male (%) | Baseline sCr (mg/dL)[b] | Settings |
|---|---|---|---|---|---|---|---|---|---|---|
| Tasanarong A[23] | 2013 | Thailand | prospective | an increase of sCr above 0.3mg/dL or 1.5 times within 48 h | 130 | 16 | CIN:70±10; non-CIN:72±7 | 100 (77) | CIN:2.00±0.60; non-CIN:1.40 ±0.40 | undergoing CAG/PCI with eGFR ≤60 ml/min per 1.73m$^2$ (except CKD 5) |
| Shukla AN [24] | 2017 | India | prospective | an increase of sCr above 0.5mg/dL or over 25% within 48h | 253 | 31 | 56.54±10.04 | 206 (81) | CIN:2.26±1.43; non-CIN:NR | undergoing CAG/PCI |
| Lacquaniti A [25] | 2013 | Italy | prospective | an increase of sCr above 0.5mg/dL or over 25% | 60 | 23 | men:57.7±11.3; women:60.6±12 | 30 (50) | 1.40±0.49 | undergoing CM enhanced CT/MRI with CKD (30≤GFR ≤60 ml/min) |
| Liao B[26] | 2019 | China | prospective | an increase of sCr above 0.5mg/dL or over 25% within 72 h | 240 | 25 | 60.92±6.38 | 128 (53) | CIN:0.77±0.13; non-CIN:0.74 ±0.09 | undergoing PCI |
| Briguori C [27] | 2010 | Italy | prospective | an increase of sCr above 0.3mg/dL at 48h | 410 | 34 | 70±9 | 344 (84) | 1.64(1.51– 1.90) | CAG/PAG/angioplasty procedure with CKD(eGFR ≤60 ml/min per 1.73m$^2$) |
| Budano C [28] | 2019 | Italy | prospective | an increase of sCr above 0.3 mg/dL at 48h or over 50% in 7 days | 713 | 47 | 66±11 | 520 (73) | 1.09±0.40 | undergoing CAG |
| Quintavalle C[29] | 2015 | Italy | prospective | an increase of sCr above 0.3mg/dL at 48 h | 458 | 64 | CIN:74±9; non-CIN:75±8 | 302 (66) | CIN:2.09 (1.15–5.32); non-CIN:1.93 (0.91–4.78) | undergoing CAG/PAG/ angioplasty procedure with eGFR ≤30 ml/min per 1.73m$^2$ or Mehran risk score≥11 |
| Souza DF [30] | 2015 | Brazil | prospective | an increase of sCr above 0.3mg/dL at 48 h | 125 | 22 | CIN:60±10.8; non-CIN:62.5 ±10.3 | 63 (50) | CIN:0.73±0.10; non-CIN:0.81 ±0.10 | undergoing CAG |
| Cecchi E[31] | 2017 | Italy | prospective | an increase of sCr above 0.5mg/dL or over 25% within 48h | 43 | 7 | 67.3±9.6 | 31 (72) | 0.85±0.17 | undergoing PCI |
| Ribichini F [32] | 2012 | Italy | prospective | an increase of sCr 0.3– 0.5mg/dL or over 25% within 48h | 166 | 30 | CIN:75 (64.3– 79.8); non CIN:72.5(63.0– 81.3) | 120 (72) | CIN:1.0 (0.77– 1.50); non-CIN:1.02 (0.90–1.38) | undergoing CA/angioplasty |
| Kim GS[33] | 2015 | Korea | retrospective | an increase of sCr above 0.5mg/dL or over 25% within 48h | 240 | 28 | 66.8±11.3 | 194 (81) | 1.20±0.60 | undergoing PTA with intermittent claudication or critical limb ischemia |
| Li H[34] | 2018 | China | prospective | an increase of sCr above 0.5mg/dL or over 25% within 72 h | 202 | 30 | 59.95±10.56 | 165 (82) | CIN:1.09 (0.99–1.27); non-CIN:1.08 (0.96–1.22) | undergoing PCI |
| Torregrosa I [35] | 2012 | Spain | prospective | an increase of sCr over 50% | 89 | 12 | CIN:73±9; non-CIN:61±13 | 67 (75) | CIN:1.20±0.30; non-CIN:0.94 ±0.22 | undergoing CAG in ICU |
| Kato K[36] | 2008 | Japan | prospective | an increase of sCr above 0.5mg/dL or over 25% within 48h | 87 | 18 | 67±11 | 62 (71) | CIN:1.05±0.28; non-CIN:1.02 ±0.18 | undergoing cardiac catheterization with/without PCI in CCU or ICU |
| Ning L[37] | 2018 | China | prospective | an increase of sCr over 50% | 168 | 20 | 66.7±3.6 | 116 (69) | CIN:0.89±0.09; non-CIN:0.96 ±0.07 | undergoing PCI |
| LIU XL[38] | 2012 | China | prospective | an increase of sCr above 0.5mg/dL or over 25% within 48 h | 311 | 39 | CIN:63.2±10.5; non-CIN:58.4 ±9.3 | 198 (64) | CIN:1.12±0.28; non-CIN:1.07 ±0.22 | undergoing CAG/PCI with mild or moderate CKD |
| Connolly M [39] | 2018 | UK | prospective | an increase of sCr above 0.3mg/dL or over 50% within 48 h | 301 | 28 | CIN:69.9±10.1; non-CIN:73.9 ±8.0 | 170 (56) | CIN:2.41±1.89; non-CIN:1.42 ±0.44 | undergoing CAG with CKD (GFR ≤60 mls/min) |

*(Continued)*

**Table 2.** (Continued)

| First Author | Year | Location | Study design | CIN definition | No. of patient | No. of CIN | Mean age[a] | Male (%) | Baseline sCr (mg/dL)[b] | Settings |
|---|---|---|---|---|---|---|---|---|---|---|
| Khatami MR [40] | 2015 | Iran | prospective | an increase of sCr above 0.3mg/dL at 48 h | 121 | 7 | 60±10.8 | 71 (59) | 0.90±0.20 | undergoing CAG |
| Padhy M[41] | 2014 | India | nested case control | an increase of sCr above 0.5mg/dL or over 25% within 48–72 h | 60 | 30 | CIN:57.63±7.36; non-CIN:54.17 ±9.35 | 44 (73) | CIN:0.86±0.24; non-CIN:0.82 ±0.19 | undergoing PCI |
| Wang M[42] | 2016 | China | prospective | an increase of sCr above 0.5mg/dL or over 25% within 72 h | 300 | 29 | 63.47±9.92 | 179 (60) | CIN:0.87±0.16; non-CIN:0.91 ±0.12 | undergoing CAG |
| Peng L[43] | 2015 | China | prospective | an increase of sCr above 0.5mg/dL or over 25% within 48h | 196 | 29 | 70.4±11.3 | 134 (68) | CIN:0.96±0.30; non-CIN:1.05 ±0.39 | undergoing PCI |
| Xu Q[44] | 2017 | China | prospective | an increase of sCr above 0.5mg/dL or over 25% within 48–72 h or a rise in cystatin C over 25% within 3 days | 213 | 52 | 52.07±14.52 | 164 (77) | CIN:0.86±0.41; non-CIN:0.81 ±0.23 | undergoing angiography |
| Alharazy SM [45] | 2014 | Malaysia | prospective | an increase of sCr over 25% within 48 h | 100 | 11 | 60.4±8.3 | 79 (79) | CIN:1.43±0.98; non-CIN:1.44 ±0.62 | undergoing CAG with CKD (stage 2–4) |
| Li S(a)[46] | 2015 | China | prospective | an increase of sCr above 0.5mg/dL or over 25% within 48h | 424 | 52 | CIN:63.5±10.8; non CIN:65.4 ±10.4 | 244 (58) | CIN:0.84±0.07; non-CIN:0.83 ±0.10 | undergoing 320-slice CCTA |
| Li S(b)[47] | 2015 | China | prospective | an increase of sCr above 0.5mg/dL or over 25% within 48h | 580 | 57 | CIN:67.2±9.4; non CIN:62.6± 10.9 | 328 (57) | CIN:0.94±0.06; non-CIN:0.93 ±0.09 | undergoing 320-slice CCTA |
| Nozue T[48] | 2010 | Japan | prospective | an increase of sCr above 0.5mg/dL or over 25% within 48–72 h | 96 | 5 | 70±10 | 73 (76) | 1.00±0.30 | undergoing PCI |
| Wang L[49] | 2014 | China | prospective | an increase of sCr above 0.5mg/dL or over 25% within 72 h | 42 | 14 | CIN:60.2±9.5; non-CIN:60.6 ±8.1 | 23 (55) | CIN:0.93±0.21; non-CIN:1.04 ±0.21 | undergoing CAG or PCI |
| Ling W[50] | 2008 | China | prospective | an increase of sCr above 0.5mg/dL or over 25% within 48–72 h | 40 | 13 | CIN:66.3±9.9; non-CIN:68.62 ±10.6 | 24 (60) | CIN:0.72±0.29; non-CIN:0.88 ±0.26 | undergoing CAG |
| Zhang WF [51] | 2017 | China | prospective | an increase of sCr above 0.3mg/dL or over 50% within 48h | 1071 | 25 | 64.8±10.2 | 713 (67) | 0.79 (0.67– 0.94) | undergoing CAG or PCI |
| Valette X [52] | 2013 | France | prospective | an increase of sCr above 0.3mg/dL or over 50% within 72 h or <0.5 ml/ kg/h of UO criteria over 6h | 90 | 30 | 60(47–67) | 74 (82) | CIN:0.85 (0.61–1.26); non-CIN:0.65 (0.47–0.81) | undergoing imaging with CM administration (angiography and CT) in ICU |
| You W[53] | 2016 | China | prospective | an increase of sCr above 0.5mg/dL or over 25% within 48–72 h | 506 | 47 | CIN:65.3±10.9; non-CIN:64.2 ±10.5 | 319 (63) | CIN:0.83±0.33; non-CIN:0.84 ±0.26 | undergoing CAG or PCI |

[a] mean age ± standard deviation or median(interquartile range)

[b] mean sCr ±standard deviation or median(interquartile range). sCr, serum creatinine; CIN, contrast-induced nephropathy; CKD, chronic kidney disease; CAG, coronary angiography; PCI, percutaneous coronary intervention; CM, contrast media; CT, computed tomography; MRI, magnetic resonance imaging; PAG, peripheral angiography; PTA, percutaneous transluminal angioplasty; eGFR, estimated glomerular filtration rate; ICU, intensive care unit; CCU, cardiac care unit; CCTA, coronary computed tomography angiography; NR, not report.

**Table 3. Diagnostic value of blood NGAL, urine NGAL and serum cystatin C to predict CIN in each study.**

| First Author | Assay | source | Time of measurement | Cutoff | TP | FP | FN | TN | Sensitivity% (95%CI) | Specificity% (95%CI) | AUROC |
|---|---|---|---|---|---|---|---|---|---|---|---|
| **Blood NGAL** | | | | | | | | | | | |
| LIU XL | ELISA | Plasma | 4h | 80 ng/ml | 20 | 53 | 19 | 219 | 96(80–100) | 77(71–82) | 0.662 |
| Connolly M | biochips | Plasma | 6h | 1337 ng/ml | 21 | 11 | 7 | 262 | 73(61–84) | 52(47–57) | 0.710 |
| Valette X | Triage NGAL test | Plasma | 24h | 113 ng/ml | 19 | 29 | 11 | 39 | 73(39–94) | 77(67–85) | 0.610 |
| Lacquaniti A | Triage NGAL test | Serum | 8h | 115 ng/ml | 23 | 5 | 0 | 32 | 51(35–68) | 81(75–85) | 0.995 |
| Liao B | ELISA | Serum | 12h | 93.93 ng/ml | 24 | 50 | 1 | 165 | 75(55–89) | 96(93–98) | 0.890 |
| Quintavalle C | ELISA | Serum | 6h | 179 ng/ml | 47 | 189 | 17 | 205 | 63(44–80) | 57(45–69) | 0.620 |
| Li H | immunoturbidimetry | Serum | 24h | 111.5 ng/ml | 26 | 64 | 4 | 108 | 100(85–100) | 86(71–95) | 0.779 |
| Padhy M | ELISA | Serum | 4h | 155.2 ng/ml | 30 | 1 | 0 | 29 | 87(69–96) | 63(55–70) | 1.000 |
| Alharazy SM | ELISA | Serum | 24h | increase of 17.7 ng/ml | 8 | 23 | 3 | 76 | 100(88–100) | 97(83–100) | 0.845 |
| **Urine NGAL** | | | | | | | | | | | |
| Tasanarong A | ELISA | urine | 6h | 117 ng/ml | 15 | 25 | 1 | 89 | 94(70–100) | 78(69–85) | 0.850 |
| Lacquaniti A | ELISA | urine | 8h | 90 ng/ml | 22 | 1 | 1 | 36 | 96(78–100) | 97(86–100) | 0.992 |
| Quintavalle C | ARCHITECT platform | urine | 6h | 20 ng/ml | 48 | 189 | 16 | 205 | 75(63–85) | 52(47–57) | 0.610 |
| Souza DF | ARCHITECT platform | urine | 2h | increase of 50% | 13 | 20 | 9 | 83 | 59(36–79) | 81(72–88) | 0.815 |
| Torregrosa I | ELISA | urine | 12h | 31.9ng/ml | 12 | 7 | 0 | 70 | 100(74–100) | 91(82–96) | 0.983 |
| Ning L | ELISA | urine | 2h | 94.4 ng/mg of creatinine | 15 | 27 | 5 | 121 | 75(51–91) | 82(75–88) | 0.632 |
| Khatami MR | ELISA | urine | 12h | 22.5 ng/ml | 5 | 48 | 2 | 66 | 71(29–96) | 58(48–67) | 0.533 |
| Wang L | ELISA | urine | 4h | 11.95 ug/L | 13 | 8 | 1 | 20 | 93(66–100) | 71(51–87) | 0.897 |
| Ling W | ELISA | urine | 24h | 9.85 ng/ml | 10 | 8 | 3 | 19 | 77(46–95) | 70(50–86) | 0.734 |
| You W | nephelometry | urine | 24h | increase of 4.65 ug/L | 44 | 90 | 3 | 369 | 94(82–99) | 80(76–84) | 0.899 |
| **Serum Cystatin C** | | | | | | | | | | | |
| Shukla AN | nephelometry | serum | 24h | increase of 10% | 31 | 49 | 0 | 173 | 100(89–100) | 78(72–83) | 0.901 |
| Briguori C | particle-enhanced nephelometric immunoassay | serum | 24h | increase of 10% | 34 | 53 | 0 | 323 | 100(90–100) | 86(82–89) | NR |
| Budano C | immunonephelometry | serum | 0h | 1.4 mg/L | 30 | 107 | 17 | 559 | 64(49–77) | 84(81–87) | 0.820 |
| Quintavalle C | NR | serum | 24h | increase of 10% | 27 | 43 | 37 | 351 | 42(30–55) | 89(86–92) | 0.660 |
| Cecchi E | nephelometry | serum | 0h | 1.18ng/ml | 6 | 8 | 1 | 28 | 86(42–100) | 78(61–90) | 0.863 |
| Ribichini F | immunonephelometry | serum | 12h | increase of 0.18 ng/ml | 14 | 69 | 16 | 67 | 47(28–66) | 49(41–58) | 0.490 |
| Kim GS | particle-enhanced nephelometric immunoassay | serum | 0 | 1.35mg/L | 21 | 42 | 7 | 170 | 75(55–89) | 80(74–85) | 0.757 |
| Torregrosa I | nephelometric immunoassay | serum | 12h | 0.8mg/L | 11 | 18 | 1 | 59 | 92(62–100) | 77(66–86) | 0.869 |
| Kato K | particle-enhanced nephelometric immunoassay | serum | NR | 1.2mg/L | 17 | 10 | 1 | 59 | 94(73–100) | 86(75–93) | 0.933 |
| Padhy M | ELISA | serum | 24h | 0.994mg/L | 30 | 1 | 0 | 29 | 100(88–100) | 97(83–100) | 1.000 |
| Wang M | NR | serum | 24h | 1.55mg/L | 24 | 6 | 5 | 265 | 83(64–94) | 98(95–99) | 0.928 |
| Peng L | particle-enhanced colorimetric immunoassay | serum | 48h | increase of 15% | 12 | 12 | 17 | 155 | 41(24–61) | 93(88–96) | 0.783 |
| Xu Q | particle-enhanced colorimetric immunoassay | serum | 48h | 1.605mg/L | 48 | 76 | 4 | 85 | 92(81–98) | 53(45–61) | 0.715 |
| Alharazy SM | particle-enhanced nephelometric immunoassay | serum | 24h | increase of 0.19mg/L | 7 | 11 | 4 | 88 | 64(31–89) | 89(81–94) | 0.800 |
| Li S (a) | immunoturbidimetric | serum | 48h | 1.61mg/dL | 52 | 0 | 0 | 372 | 100(93–100) | 100(99–100) | 1.000 |

*(Continued)*

**Table 3.** (Continued)

| First Author | Assay | source | Time of measurement | Cutoff | TP | FP | FN | TN | Sensitivity% (95%CI) | Specificity% (95%CI) | AUROC |
|---|---|---|---|---|---|---|---|---|---|---|---|
| Li S (b) | immunoturbidimetric | serum | 0 | 1.05mg/dL | 39 | 148 | 18 | 375 | 68(55–80) | 72(68–76) | 0.774 |
| Nozue T | particle-enhanced nephelometric immunoassay | serum | 0h | 1.26mg/L | 4 | 25 | 1 | 66 | 80(28–99) | 73(62–81) | 0.825 |
| Zhang WF | particle-enhanced nephelometric immunoassay | serum | 48h | increase of 15% | 20 | 178 | 5 | 868 | 80(59–93) | 83(81–85) | 0.856 |

TP, true positive; FP, false positive; FN, false negative; TN, true negative; 95%CI, 95% confidence interval; AUROC, area under receiver operating characteristics curve; ELISA, enzyme linked immunosorbent assay; NR, not report.

NGAL and serum cystatin C for contrast-induced nephropathy (CIN) was reported in 9, 10 and 18 studies, respectively.

## Quality assessment

The risk of bias and applicability concerns for the 32 included studies are shown in Fig 2. The methodological quality in all included studies was relatively high, which meant that each study satisfied at least 4 items.

Regarding the patient selection and reference standard domains, over 50% of studies were considered to have a relatively high risk of bias for heterogeneity because of the complex patients' source and unfixed definition of CIN.

## Diagnostic performance

**Blood NGAL.** For blood NGAL, the pooled sensitivity and specificity were 0.86 (95%CI: 0.69–0.95) and 0.80 (95%CI: 0.67–0.89), respectively (Fig 3). The pooled diagnostic odds ratio was 25 (95%CI: 6–108). The area under the summary receiver operating characteristic curve (AUROC) of blood NGAL was 0.90. The Q test indicated significant heterogeneity (P = 0.000); $I^2$ tests in sensitivity ($I^2$ = 87.72%) and specificity ($I^2$ = 97.43%) also demonstrated high heterogeneity. After a visual analysis of the distribution of the coupled forest plot and calculation of the correlation coefficient (0.44), the results showed that there was no significant threshold effect. The results of blood NGAL in the hierarchical summary receiver operating characteristic model were β = -0.27 (95%CI:-1.14–0.59, Z = -0.62, P = 0.538), which reflected that the shape of the SROC curve was symmetric; and λ = 3.20, which indicated that the diagnostic accuracy of blood NGAL for CIN was moderate.

**Urine NGAL.** For urine NGAL, the pooled diagnostic odds ratio (DOR) was 22 (95%CI: 8–64). The AUROC of urine NGAL was 0.89. The Q test did not indicate significant heterogeneity (P = 0.06), while $I^2$ tests still demonstrated moderate heterogeneity ($I^2$ = 52%). There was a significant threshold effect since the correlation coefficient was 0.69. The results of urine NGAL in the hierarchical summary receiver operating characteristic model were β = -0.14 (95%CI: -1.01–0.73, Z = -0.31, P = 0.753), which reflected that the shape of the SROC curve was symmetric; and λ = 3.08, which indicated that the diagnostic value of urine NGAL for CIN was moderate.

**Serum cystatin C.** For serum cystatin C, the pooled sensitivity and specificity were 0.87 (95%CI: 0.73–0.94) and 0.86 (95%CI: 0.77–0.92), respectively (Fig 4). The pooled diagnostic odds ratio was 43 (95%CI: 12–152). The AUROC of serum cystatin C was 0.93. The Q test indicated significant heterogeneity (P = 0.000); $I^2$ tests in sensitivity ($I^2$ = 90.37%) and specificity ($I^2$ = 97.01%) also demonstrated high heterogeneity. After a visual analysis of the distribution of the coupled forest plot and calculation of the correlation coefficient (0.41), the results

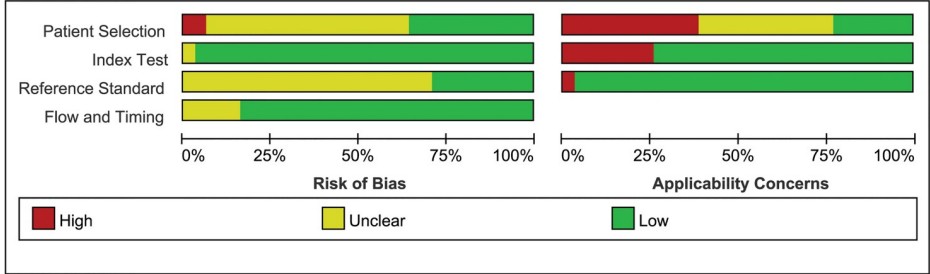

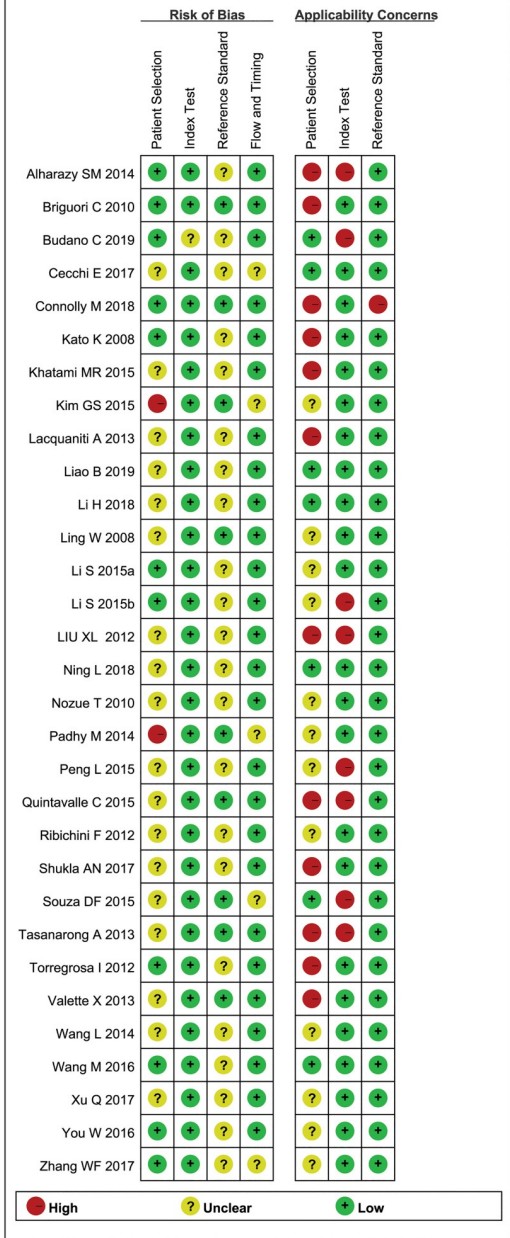

**Fig 2. The methodological quality assessment.** The methodological quality of included studies was assessed according to the Quality Assessment of Diagnostic Accuracy Studies 2 (QUADAS-2) tool.

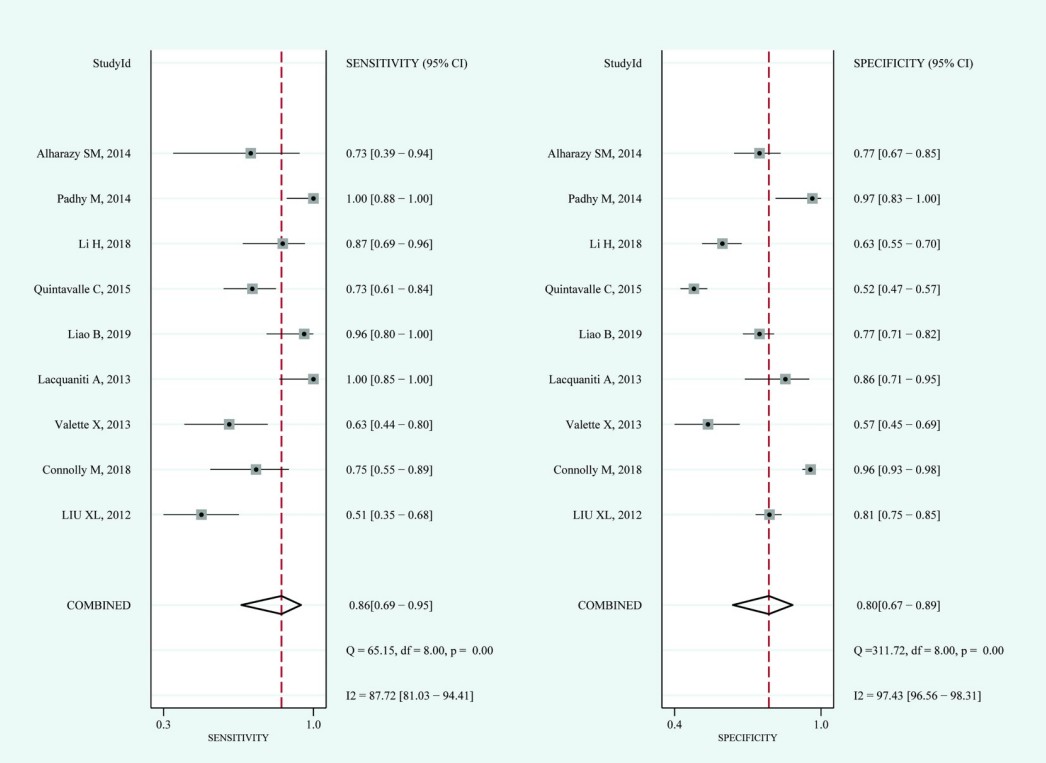

**Fig 3. Coupled forest plots for the pooled sensitivity and specificity of blood NGAL for the diagnosis of CIN.** Dots in squares represent sensitivity and specificity. Horizontal lines represent the 95% confidence interval (CI) for each included study. The pooled estimate is based on the random-effects model. Heterogeneities evaluation, $I^2$ with 95% CIs and Q are provided. Q is Cochrane heterogeneity statistic and df is the degrees of freedom.

showed that there was no significant threshold effect. The results of serum cystatin C in the hierarchical summary receiver operating characteristic model were β = -0.28 (95%CI:-0.88–0.31, Z = -0.93, P = 0.352); and λ = 3.79, which indicated that the diagnostic value of blood NGAL for CIN was moderate.

**Comparison of blood NGAL, urine NGAL and serum cystatin C.** Test comparisons of the diagnostic performance for CIN among blood NGAL, urine NGAL and serum cystatin C were conducted.

Overall, the results of the summary AUROC, DOR and λ suggested that serum cystatin C may perform better than blood NGAL and urine NGAL in diagnosing CIN. The comparison of HSROC curves is shown in Fig 5.

We compared the diagnostic accuracy of blood NGAL, urine NGAL and serum cystatin C at different cut-off times. The subgroup analysis results are shown in Table 4. The results indicated that blood NGAL may perform better than urine NGAL within 6 h after contrast media exposure; however, after 6 h, urine NGAL might be a better predictor of CIN than blood NGAL. For serum cystatin C, when measuring the level of cystatin C within 24 h after the procedure, the predictive performance was better than that at baseline.

## Sensitivity analysis and meta-regression analyses

Using Cook's distance, the sensitivity analysis showed particularly influential observations in the blood NGAL (studies from Connolly M, Padhy M), urine NGAL (study from Souza DF) and serum cystatin C (study from Li S(a)) groups.

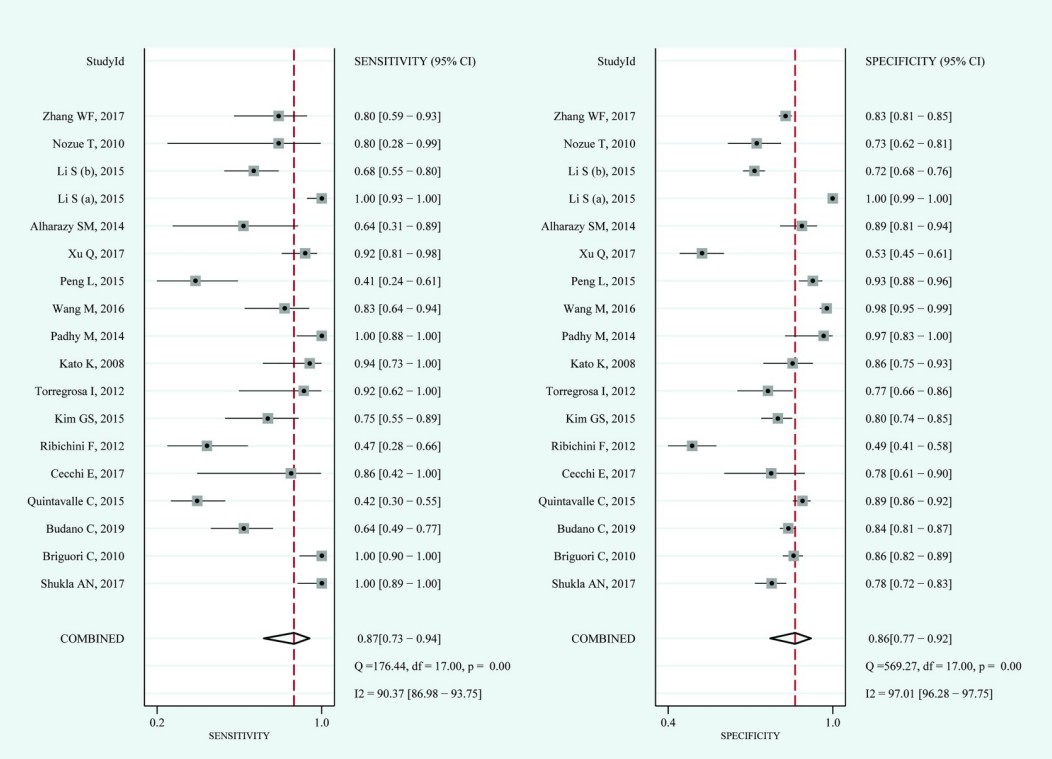

**Fig 4. Coupled forest plots for the pooled sensitivity and specificity of serum cystatin C for the diagnosis of CIN.** Dots in squares represent sensitivity and specificity. Horizontal lines represent the 95% confidence interval (CI) for each included study. The pooled estimate is based on the random-effects model. Heterogeneities evaluation, I² with 95% CIs and Q are provided. Q is Cochrane heterogeneity statistic and df is the degrees of freedom.

The meta-regression analysis results are shown in the S1 Table. Among them, the significant sources of heterogeneity were "CIN definition time", "assay" and "sample source" for the blood NGAL group; and "CIN definition time" and "location" for the urine NGAL group. In the serum cystatin C group, there was no significant source of heterogeneity initially. However, after omitting the most particular influential study, the significant source of heterogeneity came from the "assay" of detecting serum cystatin C. Other covariates were not significantly responsible for the heterogeneity between the studies.

## Publication bias

Deek's test showed that there was no significant publication bias in each group (P value = 0.08 for blood NGAL, 0.40 for urine NGAL and 0.90 for serum cystatin C) (Fig 6).

## Discussion

Owing to the constant shortcomings of serum creatinine for the early diagnosis of CIN, NGAL and cystatin C have been regarded as promising biomarkers in clinical practice. Our results suggested the following: 1) overall, the diagnostic performance of serum cystatin C is better than that of blood NGAL and urine NGAL; 2) blood and urine NGAL have similar predictive value, while the diagnostic accuracies of blood NGAL and urine NGAL were opposite within or beyond 6 h after CM exposure; and 3) serum cystatin C after CM exposure performed better in predicting CIN compared with that at baseline.

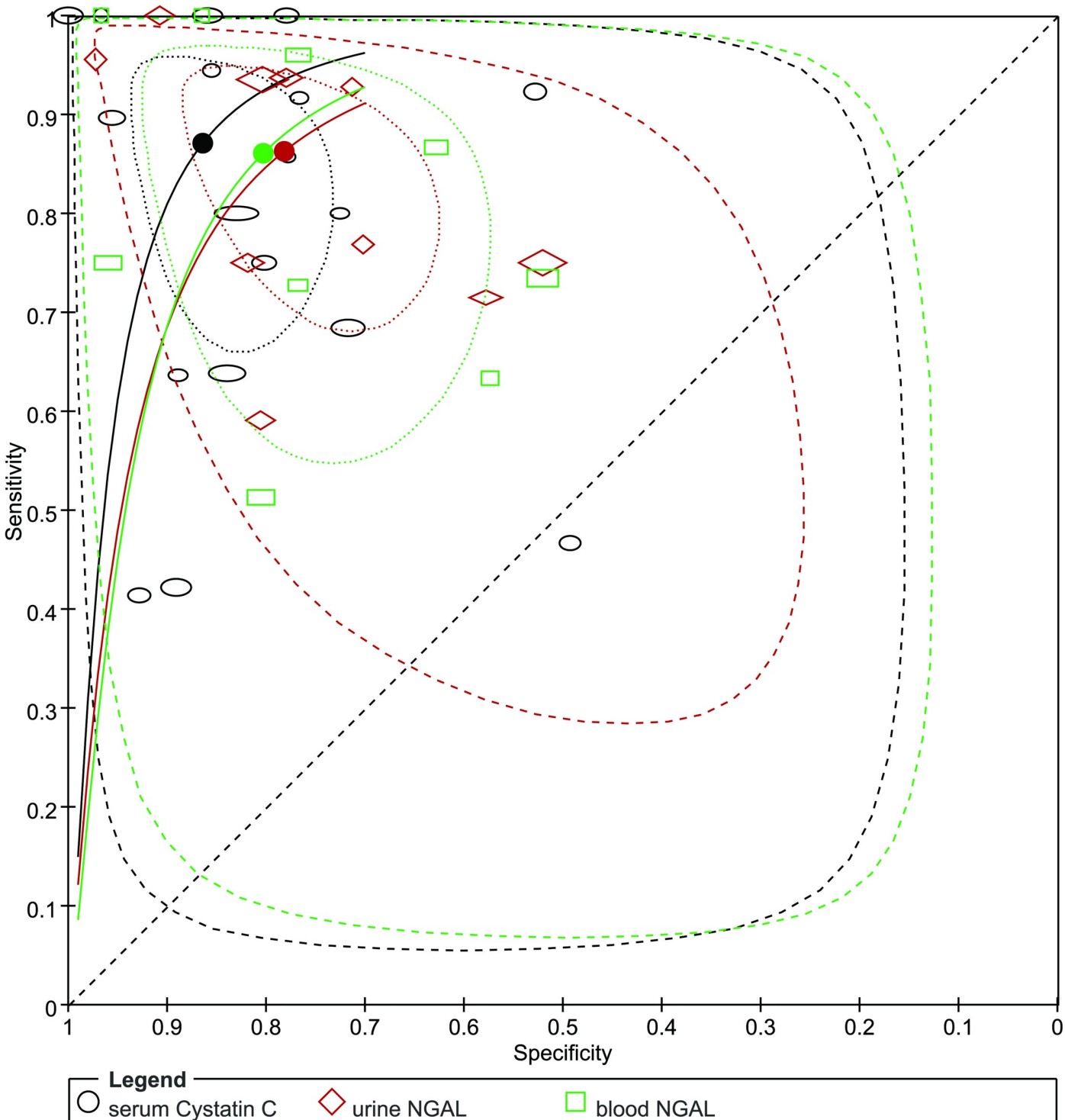

**Fig 5. Hierarchical summary receiver operating characteristic (HSROC) curve for blood NGAL, urine NGAL and serum cystatin C for the diagnosis of CIN.**
The black, green and red dots present the summary points for serum cystatin C, blood NGAL and urine NGAL respectively. The area circled by dot-dashed lines represent 95% confidence region; the area circled by dashed lines represent 95% prediction region.

**Table 4. Subgroup analysis of diagnostic performance for index tests in different measuring time.**

| Subgroups | No. of studies | Sensitivity%(95%CI)[a] | Specificity%(95%CI)[a] | DOR | AUROC | 95%CI |
|---|---|---|---|---|---|---|
| **blood NGAL** | | | | | | |
| <6h | 4 | - | - | 35 | 0.92 | 0.89–0.94 |
| >6h | 5 | - | - | 23 | 0.84 | 0.81–0.87 |
| **urine NGAL** | | - | - | | | |
| <6h | 5 | 78(64–88) | 74(62–82) | 10 | 0.83 | 0.79–0.86 |
| >6h | 5 | | | 53 | 0.94 | 0.91–0.95 |
| **serum cystatin C** | | | | | | |
| 0h(baseline) | 5 | - | - | 8 | 0.75 | 0.71–0.79 |
| <24h | 8 | 93(65–99) | 86(75–92) | 77 | 0.93 | 0.90–0.95 |

NGAL, neutrophil gelatinase-associated lipocalin; DOR, diagnostic odds ratio; AUROC, area under the summary receiver operating characteristic curve; 95% CI, 95% confidence interval. [a] Owing to the threshold effect, pooled sensitivity and specificity for some subgroups could not be calculated and presented as the absence of value.

The increase in cystatin C and NGAL levels could represent a reduction in the glomerular filtration rate and renal damage, respectively. As a low molecular weight protein, cystatin C could be freely filtered by glomeruli and completely reabsorbed and catabolized by renal tubules on normal occasions. After kidney injury, the rise of cystatin C is much earlier and superior to sCr in detecting reduced glomerular filtration rate (GFR)[54, 55]. In CIN patients, serum cystatin C was shown to peak mainly at 24 h after CM administration, which is delayed compared with the rise in serum/urine NGAL levels[56]. NGAL can be viewed as the most rapid indicator after renal tubular injuries. After iodine toxicity injury occurs in tubular cells, they secrete more NGAL than normal in response to nephrotoxic or ischemic stimuli, and the reabsorption ability of the proximal tubule is decreased. Both mechanisms contribute to the rise in serum/urine NGAL levels[57, 58]. However, NGAL can be secreted by other tissues and

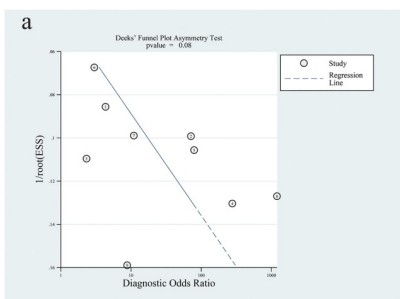
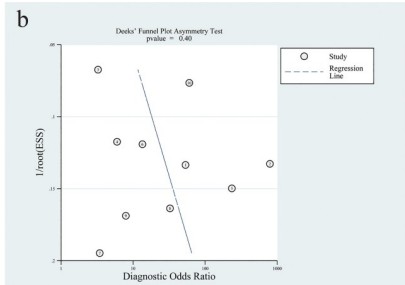
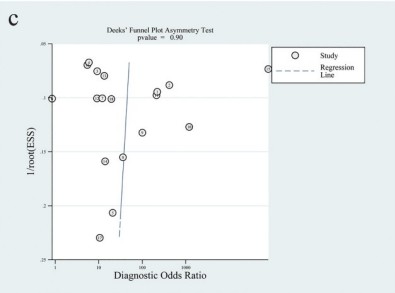

**Fig 6.** Deek's funnel plot asymmetry test for publication bias of blood NGAL(a), urine NGAL(b) and serum cystatin C (c). There was no considerable publication heterogeneity in each group.

activated by neutrophils as an acute-phase protein, which constitutes important confounders. Cecchi E et al. [31] demonstrated that serum cystatin C was associated with serum creatinine and the occurrence of CIN in patients undergoing percutaneous coronary invasive procedures (PCIPs). The rise in NGAL may suggest injury not only from the kidney but also from acute/chronic inflammation, especially in patients in intensive care settings. Singer E et al. [58] also indicated that NGAL would not be accurate enough in predicting AKI in patients with nonrenal diseases. In our results, owing to the threshold effect, we could not directly compare the pooled sensitivity and specificity between urine NGAL and serum cystatin C, while the operating points of serum NGAL and serum cystatin C were similar. Nevertheless, the summary ROC indicated that the diagnostic performance of serum cystatin C was the most valuable compared with the other two indicators. It could be interpreted that serum cystatin C, regardless of the baseline level or increases after CM exposure, could be a good predictive indicator for CIN, while NGAL is more likely to be influenced by other factors.

Several studies indicated that the rise in urine NGAL occurs a few hours later than that of blood NGAL[56, 59, 60]. Bachorzewska-Gajewska H et al.[56] demonstrated that serum and urine NGAL significantly increased at 2 and 4 hours after CM exposure, respectively, and Malyszko J et al.[60] also found that the peak of serum and urine NGAL was at 4 and 8 hours in patients undergoing cardiac catheterization. However, studies from Lacquaniti A et al.[25] and Quintavalle C et al.[29] reported that serum and urine NGAL have similar value in predicting the incidence of CIN, and our summary estimates also confirmed this view. We further investigated the diagnostic performance of serum/urine NGAL in different phases. The results indicated that blood NGAL performed well in the early phase (within 6 hours after the procedure), while the diagnostic performance of urinary NGAL was better than that of blood NGAL beyond 6 hours, which conformed to the time-course change of NGAL in serum/urine.

People with high-risk factors, such as chronic kidney disease, diabetes mellitus, dehydration, poor cardiac function, advanced age, anemia, and contrast media volume, are more likely to develop CIN[61, 62]. Among them, pre-existing CKD is the most important risk factor for CIN, and the level of serum cystatin C is higher in patients with insufficient kidney function than in the normal population[63–65]. Thus, a high level of baseline cystatin C could be seen as a predictor of high-risk populations for CIN. However, regarding the diagnostic performance of CIN, the increase in cystatin C after CM administration is better and more accurate than that at baseline.

According to those results and analyses, we proposed that it seems reasonable to combine serum cystatin C, blood NGAL and urine NGAL for diagnosing CIN. Cystatin C and NGAL have their benefits and limitations as early predictors. In clinical practice, desirable biomarkers should be sensitive and convenient to monitor in order to supply timely support; however, it should also avoid the overdiagnosis pitfall. Furthermore, it is difficult for a single marker to supply functional and damage information at the same time[5, 66]. The instant renal injury and decreased renal filtration rate can be reflected by the variation in NGAL and cystatin C, respectively, while the possibility that nonrenal factors affect CIN diagnosis would be reduced. However, care must be taken in combining biomarkers, and further investigation is needed before application in clinical practice.

There was moderate or high heterogeneity in each group since the designs and result interpretation were not standard across studies. To explore the source of heterogeneity, we further conducted subgroup and meta-regression analyses for blood/urine NGAL and serum cystatin C. First, there was no significant difference between CKD patients and other populations. We chose CKD patients as a high-risk population because other risk factors were complex and confounded. NGAL and cystatin C could be applied in different populations. Second, it is evident that diverse CIN definitions hamper the comparison across studies. According to the

diagnostic criteria from European Society of Urogenital Radiology (ESUR)[67] and Acute Kidney Injury Network (AKIN) [68], the endpoints of CIN are absolute increase of sCr of 0.5mg/dL and 0.3mg/dL or relative increase of sCr of 25% and 50% respectively. Meanwhile, the time limits are also different, within 72h and 48h separately. Based on our results, the cut-off value of the CIN definition was not responsible for the heterogeneity, but timepoint significantly influenced the diagnostic performance of NGAL. Third, when summarizing estimates, the comparability of assays for individual biomarkers should be taken into account. Assays applied in blood NGAL and cystatin C were also significant sources of heterogeneity. For NGAL, the concentrations were significantly different when using different methods[69], and the concentration of NGAL was not equivalent in plasma and serum[70]. There were also discrepancies in the diagnostic performance of cystatin C in different assays relating to the source of antibodies or different instruments[71–74]. Fourth, care should be taken in explaining the result that the diagnostic accuracy of urine NGAL was influenced by race/nationality. Only a study from Brazil[30] was responsible for the heterogeneity in the urine NGAL group. However, there was no significant difference in the diagnostic accuracy of NGAL/cystatin C between European and Asian nationalities (the results are not listed in the S1 Table).

The strength of our study is that we extensively collected studies from different countries and locations and utilized available information regarding the performance of NGAL and cystatin C in predicting CIN. Unfortunately, there are still some limitations to our study. First, we did not provide a cut-off value for separate index tests. Second, the diagnostic accuracy for the combination of cystatin C and NGAL needs further investigation. Third, the designs of the included studies were totally different and complex. Even if we enforced strict inclusion criteria and set covariates for meta-regression analysis in advance, there were still sources of heterogeneity we cannot completely explain.

In conclusion, both NGAL and cystatin C can serve as early diagnostic indicators of CIN. The combination of NGAL and cystatin C is likely to provide more diagnostic information, but more evidence is still needed.

## Supporting information

**S1 Checklist. PRISMA 2009 checklist.**
(DOCX)

**S1 Table. Meta-regression analyses for potential sources of heterogeneity from each group.**
(DOCX)

## Author Contributions

**Conceptualization:** Yi He.

**Data curation:** Yi He, Yunzhen Deng.

**Formal analysis:** Yi He.

**Funding acquisition:** Yi He.

**Investigation:** Yi He, Kaiting Zhuang.

**Methodology:** Yi He, Jing Xi.

**Project administration:** Yi He.

**Resources:** Yi He, Siyao Li.

**Software:** Yi He.

**Supervision:** Yi He, Junxiang Chen.

**Validation:** Yi He.

**Visualization:** Yi He.

**Writing – original draft:** Yi He.

**Writing – review & editing:** Yi He.

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
