## [Decision Letter · Decision Letter 0]

2 Mar 2020

PONE-D-19-34745

Predictive Value of Cystatin C and Neutrophil Gelatinase-associated Lipocalin in Contrast-induced Nephropathy: A Systematic Review and Meta-analysis

PLOS ONE

Dear Dr Chen,

Thank you for submitting your manuscript to PLOS ONE. After careful consideration, we feel that it has merit but does not fully meet PLOS ONE’s publication criteria as it currently stands. Therefore, we invite you to submit a revised version of the manuscript that addresses the minor points raised during the review process by reviewer 1.

We would appreciate receiving your revised manuscript by Apr 16 2020 11:59PM. To enhance the reproducibility of your results, we recommend that if applicable you deposit your laboratory protocols in protocols.io, where a protocol can be assigned its own identifier (DOI) such that it can be cited independently in the future. For instructions see: http://journals.plos.org/plosone/s/submission-guidelines#loc-laboratory-protocols

We look forward to receiving your revised manuscript.

Kind regards,

Emmanuel A Burdmann

Academic Editor

PLOS ONE

Journal Requirements:

2. To comply with the items on the PRISMA checklist, please your abstract using subheadings.

3. We note that you have reported significance probabilities of 0 in places. Since p=0 is not strictly possible, please correct this to a more appropriate limit, eg 'p<0.0001'.

'The funders had no role in study design, data collection and analysis, decision to publish, or preparation of the manuscript.'

Please provide an amended Funding Statement that declares *all* the funding or sources of support received during this specific study (whether external or internal to your organization) as detailed online in our guide for authors at http://journals.plos.org/plosone/s/submit-now Please state what role the funders took in the study.  If any authors received a salary from any of your funders, please state which authors and which funder. If the funders had no role, please state: "The funders had no role in study design, data collection and analysis, decision to publish, or preparation of the manuscript."

5. Please include a copy of Table 5 which you refer to in your text on page 22.

6. Please include captions for your Supporting Information files at the end of your manuscript, and update any in-text citations to match accordingly. Please see our Supporting Information guidelines for more information: http://journals.plos.org/plosone/s/supporting-information

Reviewers' comments:

Reviewer's Responses to Questions

**Comments to the Author**

1. Is the manuscript technically sound, and do the data support the conclusions?

Reviewer #1: Yes

Reviewer #2: Yes

2. Has the statistical analysis been performed appropriately and rigorously? 

Reviewer #1: Yes

Reviewer #2: Yes

3. Have the authors made all data underlying the findings in their manuscript fully available?

Reviewer #1: Yes

Reviewer #2: Yes

4. Is the manuscript presented in an intelligible fashion and written in standard English?

Reviewer #1: Yes

Reviewer #2: Yes

5. Review Comments to the Author

Reviewer #1: Page 3 line 58, please select decisions or interventions, not both.

Page 4 line 67. I suggest to add: "and catabolized by the tubular cells".

Page 4 lines 81 and 82. Systematic instead systemic.

Page 5 line 97, the same in the title of Table 1.

Page 6 line 120, pleaspaid atention to the wording.

Page 6 line 122, in "patient characteristic" comorbidities are not mentioned being relevant in CIN. However, were included in the analysis. Please correct this inconsistency.

Page 12 Table 3. Paid atention to table design. When name and last name of first author were included the line is double but stays single in the remaining columns.

Page 19 line 307, I found confusing the sentence "After kidney injury, the rise in serum cystatin C levels can be used to detect even a minor glomerular filtration rate (GFR) reduction". Please clarify.

Page 22 line 362, ESUR definition and AKIN definition are mixed. Relative increase of SCr and time limit are crossed.

There are many references there in plain text. Please check for abbreviations.

Reviewer #2: The systematic review and meta-analysis performed by Yi et al addresses an important and practical issue, i.e, the diagnostic performance of renal biomarkers, NGAL AND CYSTATIN C, in contrast induced nephropathy. A careful and detailed statistical analysis was performed pertaining the appropriate literature published until november 2019. They concluded that both NGal and especially Cystatin C, may serve as early diagnostic indicators of CIN.

6. PLOS authors have the option to publish the peer review history of their article (what does this mean?). If published, this will include your full peer review and any attached files.

Reviewer #1: Yes: Raúl Lombardi

Reviewer #2: No

---

## [Author Response · Author response to Decision Letter 0]

5 Mar 2020

Thanks a lot for your insightful comments and suggestions. Those constructive comments have helped us a lot to improve the quality of our work.

---

## [Decision Letter · Decision Letter 1]

12 Mar 2020

Predictive value of cystatin C and neutrophil gelatinase-associated lipocalin in contrast-induced nephropathy: a systematic review and meta-analysis

PONE-D-19-34745R1

Dear Dr. Chen,

We are pleased to inform you that your manuscript has been judged scientifically suitable for publication and will be formally accepted for publication once it complies with all outstanding technical requirements.

With kind regards,

Emmanuel A Burdmann

Section Editor

PLOS ONE

Additional Editor Comments (optional):

Reviewers' comments:

Reviewer's Responses to Questions

**Comments to the Author**

1. If the authors have adequately addressed your comments raised in a previous round of review and you feel that this manuscript is now acceptable for publication, you may indicate that here to bypass the “Comments to the Author” section, enter your conflict of interest statement in the “Confidential to Editor” section, and submit your "Accept" recommendation.

Reviewer #1: All comments have been addressed

2. Is the manuscript technically sound, and do the data support the conclusions?

Reviewer #1: Yes

3. Has the statistical analysis been performed appropriately and rigorously? 

Reviewer #1: Yes

4. Have the authors made all data underlying the findings in their manuscript fully available?

Reviewer #1: Yes

5. Is the manuscript presented in an intelligible fashion and written in standard English?

Reviewer #1: Yes

6. Review Comments to the Author

Reviewer #1: (No Response)

7. PLOS authors have the option to publish the peer review history of their article (what does this mean?). If published, this will include your full peer review and any attached files.

Reviewer #1: Yes: Raul Lombardi